# L4GM: Large 4D Gaussian Reconstruction Model

Jiawei Ren[1,5]    Kevin Xie[1,2]    Ashkan Mirzaei[1,2]    Hanxue Liang[1,3]
Xiaohui Zeng[1,2]    Karsten Kreis[1]    Ziwei Liu[5]    Antonio Torralba[4]
Sanja Fidler[1,2]    Seung Wook Kim[1]    Huan Ling[1,2]

[1]NVIDIA    [2]University of Toronto    [3] University of Cambridge    [4]MIT    [5]S-Lab, Nanyang Technological University

*Project page:* `https://research.nvidia.com/labs/toronto-ai/l4gm`

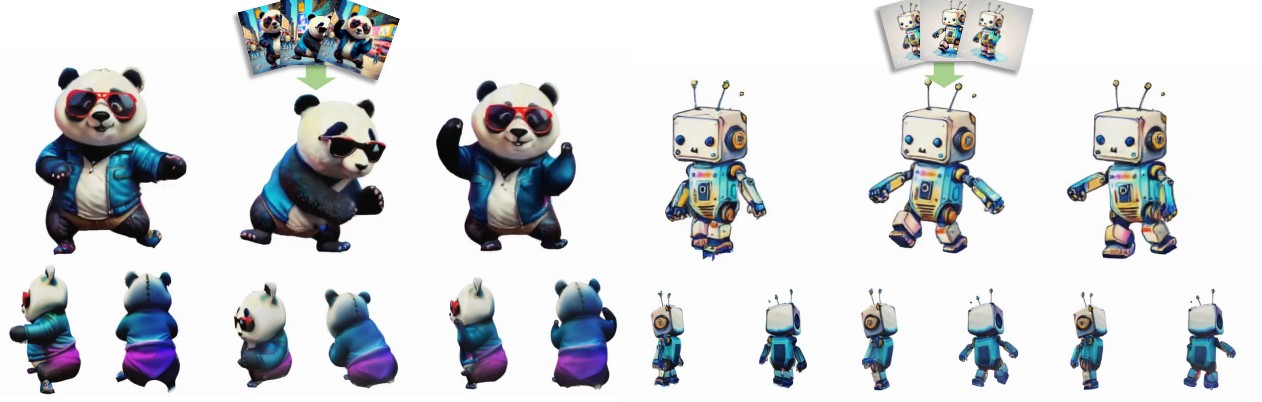

Figure 1: **L4GM** generates 4D objects from in-the-wild input videos.

## Abstract

We present **L4GM**, the first 4D Large Reconstruction Model that produces animated objects from a single-view video input – in a single feed-forward pass that takes only a second. Key to our success is a novel dataset of multiview videos containing curated, rendered animated objects from *Objaverse*. This dataset depicts 44K diverse objects with 110K animations rendered in 48 viewpoints, resulting in 12M videos with a total of 300M frames. We keep our L4GM simple for scalability and build directly on top of LGM [49], a pretrained 3D Large Reconstruction Model that outputs 3D Gaussian ellipsoids from multiview image input. L4GM outputs a per-frame 3D Gaussian Splatting representation from video frames sampled at a low fps and then upsamples the representation to a higher fps to achieve temporal smoothness. We add temporal self-attention layers to the base LGM to help it learn consistency across time, and utilize a per-timestep multiview rendering loss to train the model. The representation is upsampled to a higher framerate by training an interpolation model which produces intermediate 3D Gaussian representations. We showcase that L4GM that is only trained on synthetic data generalizes well on in-the-wild videos, producing high quality animated 3D assets.

## 1 Introduction

Animated 3D assets are essential in bringing 3D virtual worlds to life. However, these animations are time consuming to create as the procedure involves rigging and skinning of objects, and crafting keyframes of the animation – all with minimal automation in tooling. The ability to generate animated 3D assets from widely available monocular videos or simply from text would be a desirable capability for this application. This is the goal of our work. Building more advanced 4D content editing tooling, the ultimate goal of this line of research, is out of scope for this work.

38th Conference on Neural Information Processing Systems (NeurIPS 2024).

Past work on automatically generating animated 3D objects, which we refer to as 4D modeling in this paper, falls into different categories. The first line of work aims to faithfully reconstruct 4D objects from multiview video data, and oftentimes requires many views across time to achieve high quality [29, 5, 36]. Such data is expensive to collect which limits applicability. Another line of work instead relies on the power of video generative models. Most commonly, the video score distillation technique is used which optimizes a 4D representation, for example a 3D deformation field, by receiving iterative feedback from the video generative model. Score distillation is known to be fragile (sensitive to prompts), and time consuming (hours per prompt) as oftentimes many iterations are needed to achieve high quality results [47, 26, 72, 1].

Recently, a promising method emerged for the task of single-image 3D reconstruction. This method leverages large scale synthetic and real datasets to train a large transformer model, dubbed 3D Large Reconstruction Model (LRM) [20, 19], to generate 3D objects represented as neural radiance fields from a single image in a single forward pass – thus being extremely fast. We build on top of this idea to achieve fast and high quality 4D reconstruction.

We present **L4GM**, the first 4D Large Reconstruction Model (Figure 2), which aims to reconstruct a sequence of 3D Gaussians [22] from a monocular video, in a feed-forward fashion. Key to our method, is a new large-scale dataset containing 12 million multiview videos of rendered animated 3D objects from *Objaverse 1.0* [11]. Our model builds on top of LGM [49], a pre-trained 3D Large Reconstruction Model that is trained to output 3D Gaussians from multiview images. We extend it to take a sequence of frames as input and produce a 3D Gaussian representation for each frame. We add temporal self-attention layers between the frames in order to learn a temporally consistent 3D representation. We upsample the output to a higher fps by training an interpolation model that takes two consecutive 3D Gaussian representations and outputs a fixed set of in-betweens. L4GM is trained on our multiview video dataset with per-timestep image reconstruction losses by rendering the Gaussians in multiple views.

We showcase that although only trained on synthetic data, the model generalizes well to in-the-wild videos, e.g., videos generated by Sora [34] and real-world videos in AcitivityNet [12]. On the video-to-4D benchmark, we achieve state-of-the-art quality while being 100 to 1,000 times faster than other approaches. L4GM further enables fast video-to-4D generation in combination with a multiview generative model, *e.g.*, ImageDream [53].

## 2   Related Work

### 2.1   Large 3D Reconstruction Models

Reconstructing 3D representations from posed images typically requires a lengthy optimization process. Some works have proposed to greatly speed this up by training neural networks to directly learn the full reconstruction task in a way that generalizes to novel scenes [65, 55, 54, 57]. Recently, LRM [20] was among the first to utilize large-scale multiview datasets including Objaverse [11] to train a transformer-based model for NeRF reconstruction. The resulting model exhibits better generalization and higher quality reconstruction of object-centric 3D shapes from sparse posed images in a single model forward pass. Similar works have investigated changing the representation to Gaussian splatting [49, 68], introducing architectural changes to support higher resolution [61, 44], and extending the approach to 3D scenes [6, 7]. Methods such as LRM can generalize to input images supplied from sampling multiview diffusion models which enables fast 3D generation [24].

### 2.2   Video-to-4D Reconstruction

Recent works have made impressive progress in reconstructing dynamic 3D representations from multiview video inputs [28, 29]. However, for single-view video inputs, the problem of dynamic reconstruction becomes ill-posed and requires hand-crafted or data-driven priors. Good results have been achieved when targetting specific domains using shape templates [62]. Template-free methods typically require accurate depth inputs and cannot fill in portions of the object that are occluded or not visible [14, 60].

Recently, a few works have attempted to extend feed-forward generalizeable novel view synthesis to the challenging setting of dynamic monocular videos. DYST [43] introduces the synthetic 4D Dyso dataset which they use to train a transformer for generalizeable dynamic novel view synthesis

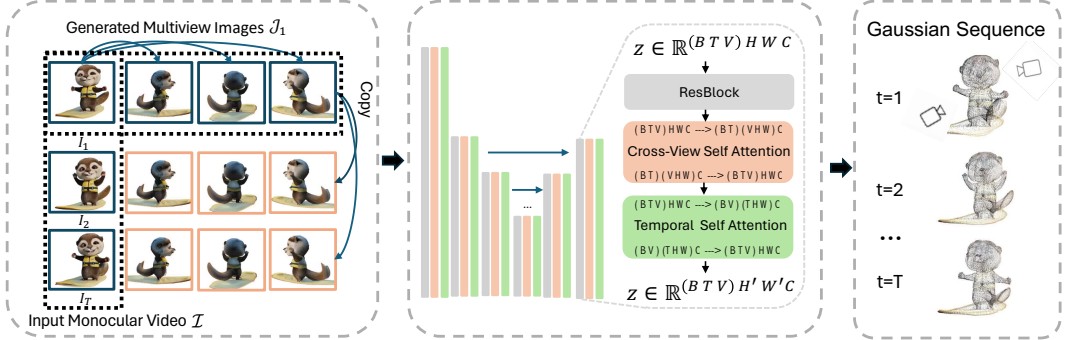

Figure 2: **L4GM.** The overall model architecture of L4GM. Our model takes a single-view video and single-time step multiview images as input, and outputs a set of 4D Gaussians. It adopts a U-Net architecture and uses cross-view self-attention for view consistency and temporal cross-time self-attention for temporal consistency.

but their model lacks an explicit 3D representation of the scene and camera. PGDVS [70] extends the generalizeable 3D NVS model GNT [54] to dynamic scenes but relies on consistent depth maps obtained through per-scene optimization. Other methods [50, 4] leverage pretraining but still require some amount of test-time finetuning to achieve acceptable novel view synthesis. Several works tackle generalizable human shape reconstruction but rely on human template meshes [31, 23].

### 2.3 Text-To-4D Generation

Dreamfusion [35] introduced the score distillation framework for text-to-3D shape generation. Such per-object optimization methods have also been extended to the 4D domain [47] where they leverage video diffusion models [46, 3, 30]. Some methods combine the guidance from video diffusion models with multiview [45] and single-image diffusion models [41] to boost 3D consistency and individual frame quality [72, 1, 26], and they utilize different forms of score distillation [25, 66, 56].

By utilizing image conditional diffusion priors such as Zero123 [27] and ImageDream [53] as guidance, this approach has also been applied to the image-conditional setting [71] and video-conditional 4D generation setting by Consistent4D [21] and DreamGaussian4D [39]. GaussianFlow [15] introduces a Gaussian dynamics based representation which supports them to add optical flow estimation as an additional regularization to SDS. STAG4D [67] and 4DGen [64] use diffusion priors to sample additional pseudo-labels from "anchor" views to expand the set of reference images for image-conditional SDS and as direct photometric reconstruction loss.

To accelerate the generation process, some works eschew the use of score distillation guidance altogether. For reconstruction from monocular video input, Efficient4D [33] and Diffusion$^2$ [63] utilize a two stage approach. In the first stage they craft schemes for sampling multiview videos conditioned on the input video. Standard optimization-based reconstruction is then used for stage 2. However, this optimization process can still take on the order of tens of minutes. In this work we directly train a feed forward 4D reconstruction model instead.

## 3 Background

Our L4GM builds on the success of single-image 3D reconstruction models [20, 19, 49], specifically the Large Multi-View Gaussian Model (LGM) [49]. LGM accepts a set of multiview images of an object and directly outputs a 3D reconstruction of the object, represented by a set of Gaussian ellipsoids $P$. Each Gaussian ellipsoid is represented by 14 parameters, including a center $\mathbf{z} \in \mathbb{R}^3$, a scaling factor $\mathbf{s} \in \mathbb{R}^3$, a quaternion rotation $\mathbf{q} \in \mathbb{R}^4$, an opacity $\alpha \in \mathbb{R}$, and a color feature $\mathbf{c} \in \mathbb{R}^3$. The multiview images $\mathcal{J} = \{J_v\}_{v=1}^V$ are taken from $V$ camera poses $\mathcal{O} = \{O_v\}_{v=1}^V$. These camera poses are encoded as image embeddings via Plücker ray embeddings and concatenated to the RGB channels of the multiview images as input to the model. They are fed through an asymmetric U-Net [42] yielding $V$ 14-channel image feature maps, where each of the pixels will be interpreted as the parameters of a 3D Gaussian.

When only a single input image is given, an image-conditional multiview diffusion model such as ImageDream [53] is first used to generate plausible completions for the missing multiview images. These generated views are then fed to LGM for reconstruction.

## 4 Our Method

Given a monocular video of a dynamic object, denoted as $\mathcal{I} = \{I_t\}_{t=1}^T$ where $T$ is video length, our objective is to rapidly reconstruct an accurate 4D representation of the object. Our approach is grounded in two conceptually simple yet impactful insights.

Our inspiration stems from video diffusion models. Recent advances in video generation have highlighted the benefits of first pretraining on image data and then extending and finetuning the model on video datasets to effectively model temporal consistency [3, 16, 59, 2]. Similarly, recognizing the scarcity of 4D data, we want to leverage a pre-trained Large Multi-View Gaussian Model (LGM) [49] that operates on images and has been extensively trained on a large-scale 3D dataset of static objects. This strategy leverages the robustness of pre-trained models to effectively train a 4D reconstruction model with limited data.

Secondly, in contrast to most existing methods [28, 29] that are required to use multiview videos for 4D reconstruction, we found that utilizing a single set of multiview images at the initial timestep is sufficient. We can obtain these multiview images easily by leveraging multiview image diffusion models to expand the first frame of the view. By adding temporal self-attention layers, our model capitalizes on the initial multiview input by propagating and adapting this information across subsequent timesteps (subsection 4.2). This approach significantly reduces the computational complexity and challenges typically associated with generating consistent multiview videos, while still enhancing the quality of the reconstruction, as our results demonstrate.

Thus, in the following, we introduce L4GM, a model that processes a monocular video to output a set of 3D Gaussians for each timestep, denoted by $\mathcal{P} = \{P_t\}_{t=1}^T$, where each $P_t$ is a set of 3D Gaussians at time $t$. L4GM is an extension of a pretrained 3D LGM, enhanced with temporal self-attention layers for dynamic modeling. We generate $V$ multiview images based on the first frame of the input monocular video. These generated views, along with the input video, are fed into L4GM to reconstruct the entire 4D sequence. We also explore further finetuning L4GM into a 4D interpolation model, allowing us to generate 4D scenes at a higher FPS than the monocular input video, offering smoother and more detailed motion dynamics within the 4D reconstructions.

### 4.1 Generate Multiview Images with ImageDream

Similar to the single-image scenario of LGM, we use ImageDream [53] to generate four orthogonal views conditioned on the initial frame $I_1$. We denote $\mathcal{J}_1$ as the set of generated multiview images taken from camera poses $\mathcal{O}$ at the initial time step $t = 1$. We would like our generated multiview images to contain three extra views that are orthogonal to the input first frame of the original video.

However, often none of the viewing angles of the generated multiview images match the input frame $I_1$. An example can be found in Appendix Figure 8. To address this, we first use the 3D LGM to reconstruct an initial set of 3D Gaussians, $P_{\text{init}}$, from the generated multiview images, and render this reconstruction from views that are orthogonal to $I_1$ as desired. We provide exact details in Appendix E.

### 4.2 Turning the 3D LGM into a 4D Reconstruction Model

**Model Architecture.** We adopt the asymmetric U-Net [42] structure from the pretrained LGM as backbone. To align with LGM's input specifications, we replicate the generated multiview images in $\mathcal{J}_1$ at $t = 1$ (except $I_1$) across all other time steps to construct a $T \times V$ grid (see left side of Figure 2 for an example). For simplicity, we assume the camera in the reference monocular video is static and only the object is moving so we also copy the camera poses $\mathcal{O}$ (except $\mathcal{O}_1$) across time steps. Similar to LGM, these poses are then embedded using Plücker ray embeddings [48]. We concatenate this camera embedding with the RGB channels of the input images. The concatenated inputs are reshaped into the format (B T V) H W C and fed into the asymmetric U-Net, where B is batch size.

As illustrated in the middle section of Fig 2, each U-Net block within L4GM consists of multiple residual blocks [18], followed by cross-view self-attention [52] layers. To maintain temporal consistency across different timestamps, we introduce a new *temporal* self-attention layer following each cross-view self-attention layer. These *temporal* self-attention layers treat the view axis V as a batch of independent videos (by transferring the view axis into the batch dimension). After processing, the data is reshaped back to its original configuration. In einops [40] notation, this process looks as:

$$\mathbf{x} = \mathtt{rearrange}(\mathbf{x}, \mathtt{(B\ T\ V)\ H\ W\ C} \rightarrow \mathtt{(B\ V)\ (T\ H\ W)\ C}) \tag{1}$$

$$\mathbf{x} = \mathbf{x} + \mathrm{TempSelfAttn}(\mathbf{x}) \tag{2}$$

$$\mathbf{x} = \mathtt{rearrange}(\mathbf{x}, \mathtt{(B\ V)\ (T\ H\ W)\ C} \rightarrow \mathtt{(B\ T\ V)\ H\ W\ C}) \tag{3}$$

where $\mathbf{x}$ is the feature, B H W C are batch size, height, width, and the number of channels.

The output of the U-Net consists of 14-channel feature maps with shape $B \times T \times V \times H_{\mathrm{out}} \times W_{\mathrm{out}} \times 14$. Each of the $1 \times 14$ units is treated as the set of parameters of a per-pixel Gaussian. We concatenate these Gaussians along the view dimension V to form a single set of Gaussians for each timestamp, resulting in $T$ sets of 3D Gaussians, $\{P_t\}_{t=1}^{T}$. This collection forms our final 4D representation.

**Loss Functions.** Besides the input camera poses $\mathcal{O}$, we select another set of camera poses $\mathcal{O}_{\mathrm{sup}}$ for multiview supervision. We train the model with a simple reconstruction objective on the video rendering of the output 4D representations from camera poses $\mathcal{O} \cup \mathcal{O}_{\mathrm{sup}}$, as detailed in Appendix B.

### 4.3 Autoregressive Reconstruction and 4D Interpolation

In practice, one may want to apply L4GM on long videos and obtain temporally smooth 4D outputs. To this end, L4GM also enables *autoregressive reconstruction* which processes videos in an autoregressive fashion, one chunk of $T$ frames after another. We additionally train a *4D Interpolation Model* to upsample the 4D representation to a higher fps. Details can be found in Section A.

**Autoregressive Reconstruction** (Figure 3, left). Our base model is designed to accept a monocular video of fixed length $T$. For long videos that exceed $T$, we partition the video into chunks of size $T$ which we process sequentially. We first apply L4GM to the initial $T$ frames to generate the first set of $T$ Gaussians. Subsequently, instead of generating multiview images for the first frame of the next chunk using a multiview diffusion model, we render the last set of Gaussians from four orthogonal angles to obtain new multiview images. These newly rendered multiview images, along with the next set of video frames, are then used to generate the next set of Gaussians. The process is repeated until all frames of a long video have been reconstructed. We empirically observe that such an autoregressive reconstruction method can be repeated more than 10 times without a significant drop in quality.

**4D Interpolation Model** (Figure 3, right). As our model does not track Gaussians across frames, directly interpolating Gaussian trajectories is not feasible [29, 58]. Hence, we develop an interpolation model that operates in 4D, fine-tuned on top of L4GM. Similar interpolation methods have been used successfully in the video generation literature [3]. As shown in Figure 3, the input to the 4D Interpolation Model consists of two sets of multiview images and the interpolation model is trained to produce additional intermediate sets of Gaussians. It leverages the weight-average of the RGB pixels between the multiview images for the newly created intermediate frames. The 4D Interpolation Model then outputs the corresponding sets of Gaussians. In practice, we insert two additional time frames.

## 5 Objaverse-4D dataset

**Dataset Collection** To collect a large-scale dataset for the 4D reconstruction task, we render all animated objects in *Objaverse 1.0* [11].

Out of 800,000 objects, only 44,000 have animations. Since each object can have multiple associated animations, we obtain a total of 110,000 animations. All of the animations are 24 fps.

The dataset consists mostly of animations on rigged characters or objects, e.g., *"a T-Rex walking"* or *"a windmill spinning"*. The motions include diverse scenarios such as dynamic locomotions, fine-grained facial expressions, and smoke effects. Interestingly, there are a considerable amount of deforming motions in the dataset, since many animations feature a fantasy style – thus, even rigid real-world objects are deformable. See Appendix Figure 7 and supplementary video for examples.

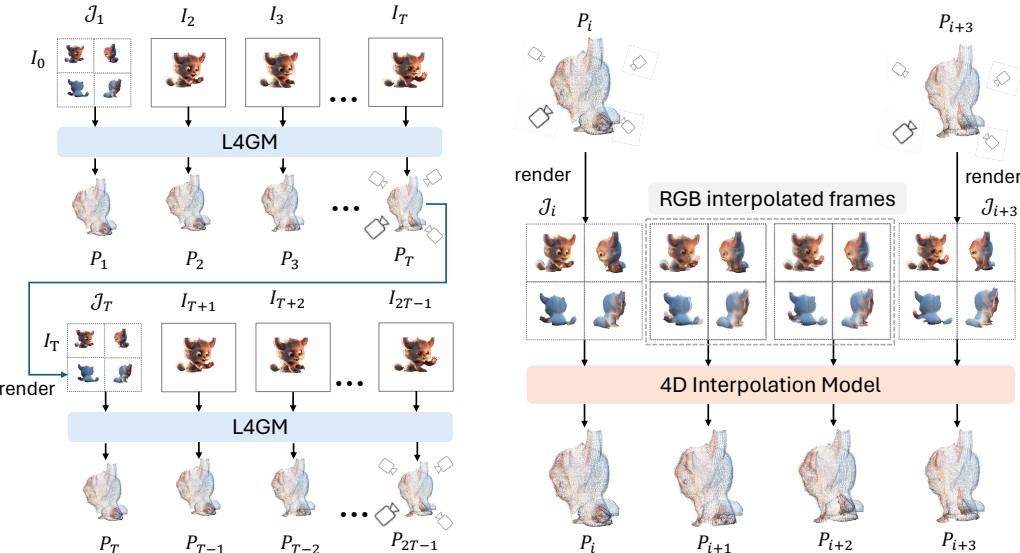

Figure 3: **Left: Autoregressive reconstruction.** We use the multiview rendering of the last Gaussian as the input to the next reconstruction. There is a one-frame overlap between two consecutive reconstructions. **Right: 4D Interpolation.** The interpolation model takes in the interpolated multiview videos rendered from the reconstruction results and outputs interpolated Gaussians.

**Dataset Rendering** Following [53, 32, 24], we adopt the assumption that the real-world monocular videos mostly have $0°$ elevation camera poses, and thus render input views for our training data accordingly. Specifically, since animations are of varying lengths, we split each animation into 1 second subclips and render each 4D object into 48 views $\times$ 1 second long clips. The views are from *1) 16 fixed cameras,* where cameras are placed at $0°$ elevation with uniformly distributed azimuths, and *2) 32 random cameras,* where cameras are placed at random elevations and azimuths. During training, we sample input camera poses $\mathcal{O}$ from the 16 *fixed* cameras and sample the supervision cameras $\mathcal{O}_{\text{sup}}$ from the 32 *random* cameras. Furthermore, following [2], we further filter out approximately 50% of the 26M videos with small motion based on optical flow magnitude, resulting in a total of 12M videos in Objaverse-4D dataset.

## 6 Experiments

### 6.1 Implementation Details

We rendered the dataset with Blender and the EEVEE engine [9]. We used fixed camera intrinsics and lighting as detailed in the appendix. For L4GM, we downsample the clips to 8 FPS and train the model for 200 epochs. In training, we set $T = 8$ and use 4 input cameras and 4 supervision cameras. During inference, we used $T = 16$, which we empirically show to work well for longer videos in section 6.3. Each forward pass through L4GM takes about 0.3 seconds, while generating sparse views requires about 2 seconds. For training the interpolation model, we use the 24 FPS clips without downsampling and fine-tune L4GM for another 100 epochs. The 4D interpolation model takes 0.065 seconds to interpolate between every two frames (see Appendix for details).

### 6.2 Comparisons to State-of-the-Art Methods

We focus our evaluations on video-to-4D reconstruction. Although L4GM can also be used for text-to-4D or image-to-4D synthesis by taking text-to-video or image-to-video outputs from video generative models as input, existing text-to-4D [26, 1, 46] and image-to-4D [71] approaches typically rely on score distillation and are orders of magnitudes slower than L4GM, preventing meaningful comparisons. Hence, we concentrate on video-to-4D. Results are presented in the following paragraphs.

**Quantitative Evaluation.** We evaluate our L4GM on the benchmark provided by Consistent4D [21]. It consists of eight dynamic 3D animations of 4 seconds in length, at 8-FPS. A video from one view is

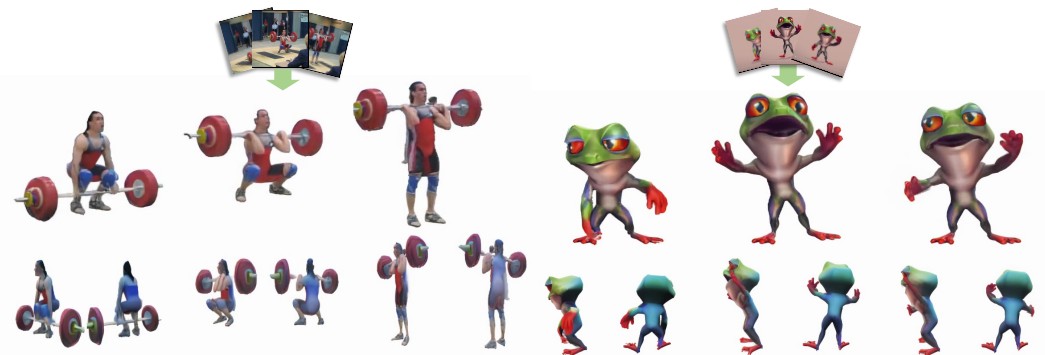

Figure 4: Qualitative results from L4GM, showcasing renderings from 4D reconstructions produced from two in-the-wild videos.

used as input and 4 videos from other viewpoints are used for evaluation. Three metrics are computed: *1)* Perceptual similarity (LPIPS) between the generated and the ground truth novel views, *2)* the CLIP [38] image similarity between the generated and the ground truth novel views, and *3)* the FVD [51] against the ground truth novel views , which measures video quality. We also report the runtime. We reconstruct the video at the original frame rate without using interpolation model. The results are shown in Table 1. L4GM outperforms existing video-to-4D generation approaches on all quality metrics by a significant margin, while being 100 to 1,000 times faster.

**Qualitative Evaluation.** Figure 4 illustrates the renderings produced by L4GM on two videos from different angles and timesteps. These examples are taken from the ActivityNet [12] and Consistent4D [21] datasets. As shown in the figure, L4GM produces high-quality, sharp renderings while exhibiting strong temporal and multiview consistency.

We compare our visual results to DG4D [39], STAG4D [67], and OpenLRM [19]. DG4D and STAG4D are optimization-based approaches

Table 1: **Quantitative results for video-to-4D.** Best is bolded. †: results from Gao et al. [15].

| Method | LPIPS↓ | CLIP↑ | FVD↓ | Time↓ |
|---|---|---|---|---|
| Consistent4D [21] | 0.16 | 0.87 | 1133.44 | 2 hr |
| 4DGen [64] | 0.13 | 0.89 | - | 1 hr |
| GaussianFlow† [15] | 0.14 | 0.91 | - | - |
| STAG4D [67] | 0.13 | 0.91 | 992.21 | 1 hr |
| DG4D† [39] | 0.16 | 0.87 | - | 10 min |
| Efficient4D [33] | 0.14 | 0.92 | - | 6 min |
| Ours | **0.12** | **0.94** | **691.87** | **3s** |

that take 10 minutes and 2 hours on 64-frame videos, respectively. OpenLRM is an opensource work reproducing LRM [20] that reconstructs 3D shapes from single-view images. We run OpenLRM on every video frame to construct a 3D sequence. We collect 24 evaluation videos from Emu [16], Sora [34], Veo [10], and ActivityNet [12], covering both generated videos and real-world videos. For our approach, we use $T = 16$ and use the interpolation model. Some qualitative comparisons are presented in Figure 5. Notably, a significant improvement from our approach is a higher 3D resolution. Optimization-based approaches use only thousands of Gaussians to keep the optimization tractable, while our feed-forward approach can easily reconstruct more than 60,000 Gaussians per frame at a dramatically faster speed.

We further conduct a user study based on qualitative comparisons, and the results are shown in Table 2. Our approach is the most favorable on all evaluation criteria including *overall quality*, *3D appearance*, *3D alignment with input video*, *motion alignment with input video*, and *motion realism*. More details about the evaluation dataset and the user study are in Appendix G.

## 6.3 Ablation Studies

We carry out a variety of ablation studies. For training models in the ablation study, we only keep animations from high-quality objects in *GObjaverse* [37], which accounts for ≈25% of the data.

**3D Pretraining.** Without 3D pre-taining, i.e. without initializing from LGM [49], our model fails to converge (using the same training recipe). The explanation is likely that without large-scale pre-training on static 3D scenes our Objaverse-4D dataset is insufficient for L4GM to not only learn temporal dynamics but also its 3D understanding from scratch. Moreover, starting from a fully random initializion may also contribute to training instabilities. Note that when reducing the model size to the "small" LGM variation [49], the model starts to converge. However, as shown in Table 6 (a), the model converges to a significantly lower PSNR in the same training epochs.

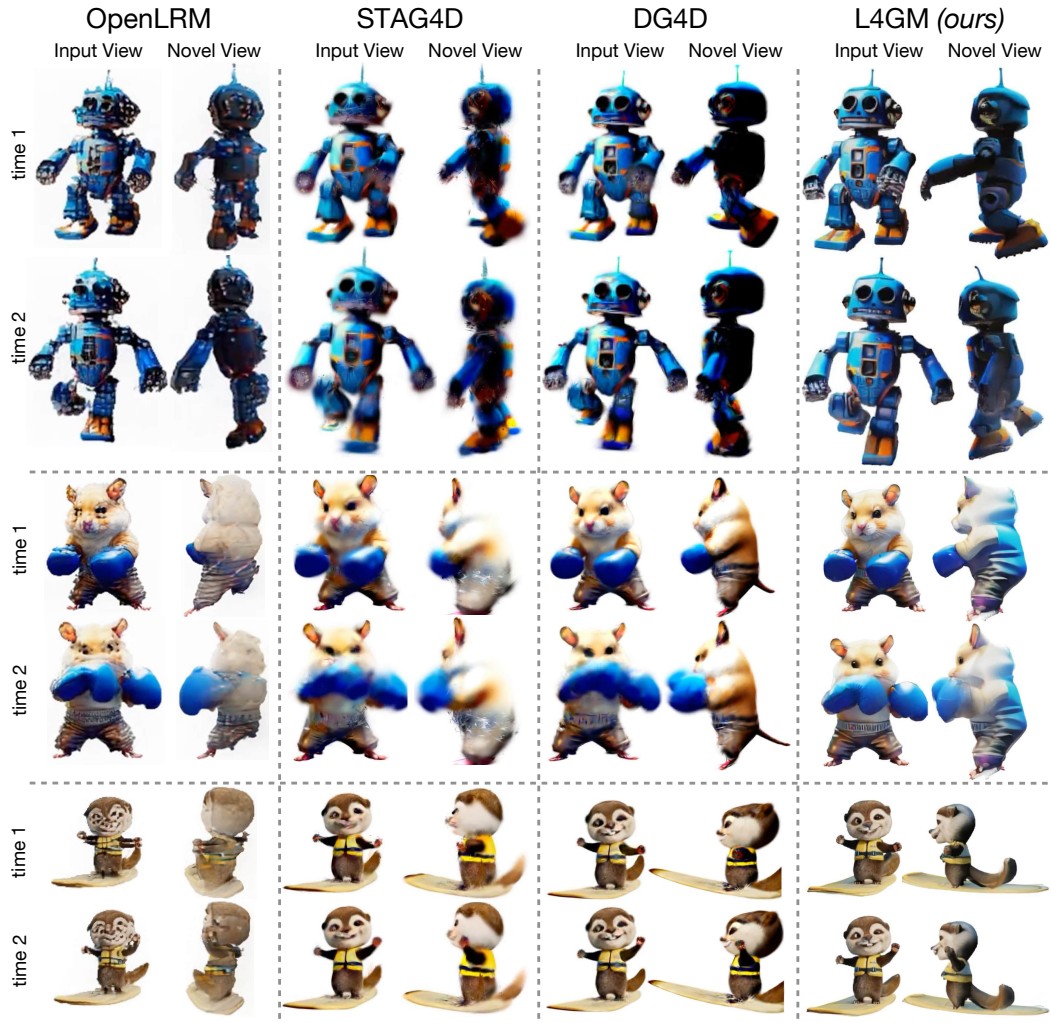

Figure 5: **Qualitative comparisons** of L4GM's results against the baselines.

**Frozen LGM.** In this experiment, we freeze the layers of the original LGM model and only train the temporal attention layers. As shown in Table 6 (b), the model improves faster at the beginning of the training but converges to a lower PSNR. This suggests that end-to-end fine-tuning of L4GM, including the non-temporal layers of the 3D LGM, is preferable for the 4D reconstruction task.

**Temporal Attention.** Without temporal attention, the model falls back into a 3D reconstruction model, while receiving asynchronous multiview inputs. The model does a surprisingly good job using only the 3D information, but it still converges to a lower PSNR, as shown in Table 6 (b). The reconstructed novel view videos contain visible flickering due to the lack of temporal modeling. Please refer to the supplementary video for a comparison.

**Deformation Field.** Since different types of deformation fields and HexPlane representations have recently been used in the text-to-4D literature [47, 26, 1, 72], we modify the model to predict a canonical 3D representation and a deformation field based on a HexPlane [5]. Concretely, we average the Gaussians as a canonical 3D representation and introduce a new decoder after the middle block in the U-Net to predict a HexPlane. The representation follows Wu et al. [58]. A detailed illustration can be found in the appendix. Although the model can successfully overfit to a single 4D data sample, it fails to learn a reasonable deformation field during large-scale training. As shown in Table 6 (b), the PSNR only slowly improves and the output is always static. This observation is very different from previous optimization-based 4D generation works [39, 26]. We speculate that SDS-based methods often rely on the smoothness of implicit representations to regularize their generations, whereas our L4GM model may have directly learned to output smooth representations from the 4D training data.

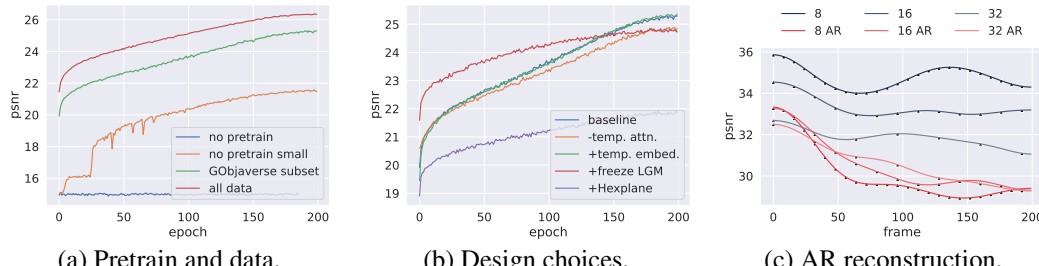

| (a) Pretrain and data. | (b) Design choices. | (c) AR reconstruction. |

Figure 6: PSNR plot. *a)* Training with different pretrain and training data. *b)* Training with different design choices. *c)* Per-frame PSNR with different video lengths $T$, AR denotes autoregressive.

Table 2: **Comparison to baselines** by user study on synthesized 4D scenes with 24 examples. Numbers are percentages. Numbers do not add up to 100; difference is due to users voting "no preference" (details in Appendix).

| L4GM *(ours)* | v.s DG4D [39] | v.s OpenLRM [19] | v.s STAG4D [67] |
|---|---|---|---|
| Overall Quality | **65.4**/25.0 | **57.9**/33.3 | **54.2**/35.0 |
| 3D Appearance | **67.1**/25.8 | **58.8**/31.7 | **55.0**/34.2 |
| 3D Alignment w. Input Video | **61.3**/26.3 | **51.3**/32.1 | **50.0**/36.7 |
| Motion Alignment w. Input Video | **61.3**/22.6 | **50.9**/29.2 | **50.4**/31.7 |
| Motion Realism | **62.1**/30.0 | **54.2**/35.0 | **50.4**/36.7 |

**Autoregressive Reconstruction.** Our model allows taking a video with a length different from the training video length. Here, we analyze the effect of the test-time video length $T$ and the number of autoregressive steps on the reconstruction quality. We use a long animation in Objaverse-4D dataset and compute the per-frame reconstruction PSNR. Results are in Table 6 (c). When the ground-truth multiviews are provided, the quality slightly drops when using longer video length. When reconstructing autoregressively, the quality decreases with more autoregressive runs. A shorter video length will start with a higher quality, but the quality drops faster than a longer video length because more self-reconstructions are required. In practice, we select $T = 16$, which offers a balanced performance for different video lengths.

**Time Embedding.** Here, we explore adding a time embedding to L4GM so that the model is aware of the ordering of the frames. Timestamps are encoded by a sinusoidal function and added to the camera embedding. However, the PSNR did not improve after adding the time embedding. We speculate that the image frames from the input video are already giving sufficient information about the temporal relation between the timesteps. Therefore, we do not train with temporal embedding to add more flexibility to the model at inference time.

**4D Interpolation.** Finally, we show a comparison of using vs. not using the 4D interpolation model on an 8-FPS video from the Consistent4D dataset in the supplementary video. Notably, the 4D interpolation model can successfully improve the framerate beyond the input video framerate.

# 7 Conclusions

We presented L4GM, the first large reconstruction model for dynamic objects. It produces dynamic sequences of sets of 3D Gaussians from a single-view video. The model leverages prior 3D knowledge from a pretrained 3D reconstruction model, and learns the temporal dynamics from a synthetic dynamic dense-view 4D dataset, Objaverse-4D dataset, which we collect. L4GM can reconstruct long videos and uses learned interpolation to achieve high framerates. We achieve orders of magnitude faster inference times than existing 4D reconstruction or text-to-4D methods. Moreover, our model generalizes well to in-the-wild real and generated videos. Our work is an early attempt at AI tooling for 4D content creation, with many challenges remaining. For example, for making this technology really useful for professionals we need to develop more advanced human-in-the-loop 4D editing capabilities.

**Broader Impact.** L4GM allows fast 4D reconstruction from in-the-wild videos, which is useful for various graphics and animation applications. However, this model should be used with an abundance of caution to prevent malicious impersonations. The model is also trained on non-commercial public datasets and is for research-only purpose.

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

# Contents

# A Autoregressive Reconstruction and 4D Interpolation Model Details

**Autoregressive Reconstruction for Longer Videos.** Our model accepts a monocular video $\mathcal{I} = \{I_t\}_{t=1}^{T}$ with a fixed length $T$ as input, where $T$ is a hyperparameter set during training (note that during inference $T$ can in principle be longer than during training). For videos longer than $T$, we can also operate our model in an autoregressive manner. Consider a long video $\{I_t\}_{t=1}^{L}$ where the frame length $L$ significantly exceeds $T$. We first apply L4GM to the initial $T$ frames to generate the first set of $T$ Gaussians $\{P_t\}_{t=1}^{T}$. Subsequently, we render the last Gaussian $P_T$ from four orthogonal angles to obtain new generated multiview images $\mathcal{J}_T = \{f(P_T, \Delta\theta)\}_{\Delta\theta \in \{0°, 90°, 180°, 270°\}}$. The rendered multiview images $\mathcal{J}_T$, along with frames $\mathcal{I} = \{I_t\}_{t=T}^{2T-1}$, are used to construct the next set of $T-1$ Gaussians $\{P_t\}_{t=T+1}^{2T-1}$. This process is repeated until all $L$ frames have been reconstructed. We find that this autoregressive reconstruction method can be repeated more than 10 times without a significant drop in quality (see Figure 3, left).

**4D Interpolation Model.** Since our model does not track Gaussians across frames, interpolating Gaussian trajectories is not feasible [29, 58]. Hence, we developed an interpolation model that operates directly within the 4D space, fine-tuning it on top of L4GM. As demonstrated in Figure 3, right side, the input to the 4D Interpolation Model consists of two sets of multiview images, $\mathcal{J}_i$ and $\mathcal{J}_{i+3}$. During training, we render $\mathcal{J}_i$ and $\mathcal{J}_{i+3}$ at 8 FPS from an asset in our dataset. During inference, $\mathcal{J}_i$ and $\mathcal{J}_{i+3}$ are rendered from reconstructed sets of gaussians $P_i$ and $P_{i+3}$. To create the $T \times V$ input grid, we insert two time steps between the input views by simply calculating the weighted-average on the RGB pixels between the multiview images $\mathcal{J}_i$ and $\mathcal{J}_{i+3}$, resulting in a total of $4 \times V$ images. The 4D Interpolation Model then outputs the corresponding four sets of Gaussians, which are supervised using the ground truth data at 24 FPS. During inference, we render the reconstruction results into multiview videos and feed them into the interpolation model to produce a sequence of Gaussians at a three times higher framerate.

# B More Implementation Details

**Dataset.** We render the dataset with the EEVEE engine in Blender [9], which requires three days with 200 GPUs. For *random* cameras, we select a random elevation from [-5°, 60°]. For the *fixed* camera setting, we rendered from 16 views equally distributed at 0° elevation looking at the origin. We set the camera radius to 1.5 to align with LGM so that we can leverage its 3D pretrain. The camera FOV is set to 49.1°. We use the lighting from the Objaverse-XL codebase[1] while fixing the strength to the mean value.

Following Blattmann et al. [2], we also employ motion filtering. Specifically, we downsample the 4D animation clips of Objaverse-4D dataset to 2-FPS and compute the optical flow magnitude averaged on videos rendered from four orthogonal fixed 0-elevation cameras. A histogram of the optical flow magnitude is shown in Table 3. We only keep the animation clips with an average optical flow magnitude larger than 0.15, resulting in 51K 4D animations with 28K different objects, 246K 1-second clips, and 12M videos rendered from 48 camera poses.

**Model.** We use the pretrained LGM model [49] released in the official code[2], which has 6 down-sampling blocks with channels [64, 128, 256, 512, 1024, 1024], 1 middle block with channel [1024], and 5 upsampling blocks with channels [1024, 1024, 512, 256, 128]. The input image size is 256x256. The number of output Gaussians for each frame is $128 \times 128 \times 4 = 65{,}536$. The output Gaussians are rendered into images at 512x512 resolution for supervision. The added temporal attention layers are the same as the cross-view attention layers, and are inserted after the same U-Net blocks.

---

[1]https://github.com/allenai/objaverse-xl
[2]https://github.com/3DTopia/LGM

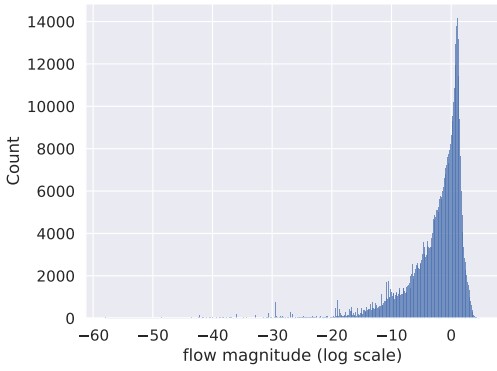 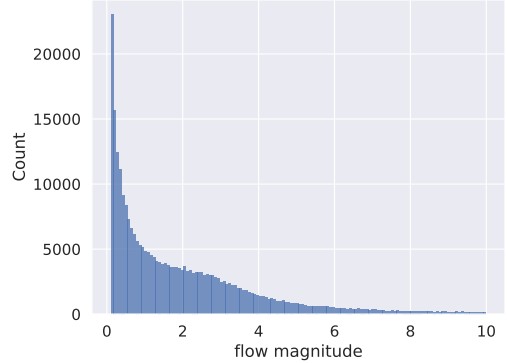

(a) Flow magnitude histogram in log scales.   (b) Flow magnitude histogram in [0.1, 10].

Table 3: Optical flow magnitude histogram on Objaverse-4D dataset.

**Loss Functions.** Following LGM, we use a combination of an LPIPS [69] loss and an MSE loss on RGB images, and an MSE loss on segmentation masks to supervise the reconstruction,

$$\mathcal{L}_{\text{RGB}} = \sum_{t=1}^{T} \sum_{O \in \mathcal{O} \cup \mathcal{O}_{\text{sup}}} ||I_t^O - f(P_t, O)||_2^2 + \lambda \mathcal{L}_{\text{LPIPS}}(I_t^O, f(P_t, O)), \tag{4}$$

$$\mathcal{L}_{\text{Mask}} = \sum_{t=1}^{T} \sum_{O \in \mathcal{O} \cup \mathcal{O}_{\text{sup}}} ||\alpha_t^O - g(P_t, O)||_2^2, \tag{5}$$

$$\mathcal{L} = \mathcal{L}_{\text{RGB}} + \mathcal{L}_{\text{Mask}}, \tag{6}$$

where $f, g$ are Gaussian volume rendering functions for RGB and alpha masks, $I_t^O$ and $\alpha_t^O$ are the image and mask rendered from camera pose $O$ at time $t$, and $\lambda$ is a loss weight.

**Training.** For training L4GM, we downsample the videos to 8 FPS and set $T = 8, V = 4$. We apply the grid distortion augmentation [49] to non-reference views to improve the model robustness. We train the model with one 8-frame clip per GPU on 128 80G A100 GPUs. In each epoch, we iterate through all animations and sample a one-second clip from it regardless of the animation length. The model is trained for 200 epochs, which takes about one day. For ablation models, we reduce the dataset size to the 25% GObjaverse subset and reduce the number of GPUs to 32 correspondingly while keeping other settings unchanged. For training the interpolation model, we fine-tune L4GM with 4 frames per GPU on 64 80G A100 GPUs for 100 epochs.

**Inference.** We test on a 16G RTX 4080 Super GPU. We set the video length to $T = 16$ Each forward pass through L4GM takes about 0.3 seconds while generating sparse views requires about 2 seconds, including ImageDream generation, LGM reconstruction, and azimuth selection. The 4D interpolation model takes 0.065 seconds to interpolate between every two frames. For example, for a 10-second 30-FPS video, we can reconstruct the input video in 15 FPS and interpolate the result to 45 FPS in 15 seconds, consisting of 2 seconds for sparse view generation, 3 seconds for reconstruction, and 10 seconds for interpolation. Video segmentation time is not included.

## C Example Training Data

We show example training data in Figure 7.

## D Full Quantitative Results on Consistent4D Benchmark

We present the detailed metric for every test sample on the Consistent4D benchmark in Table 4.

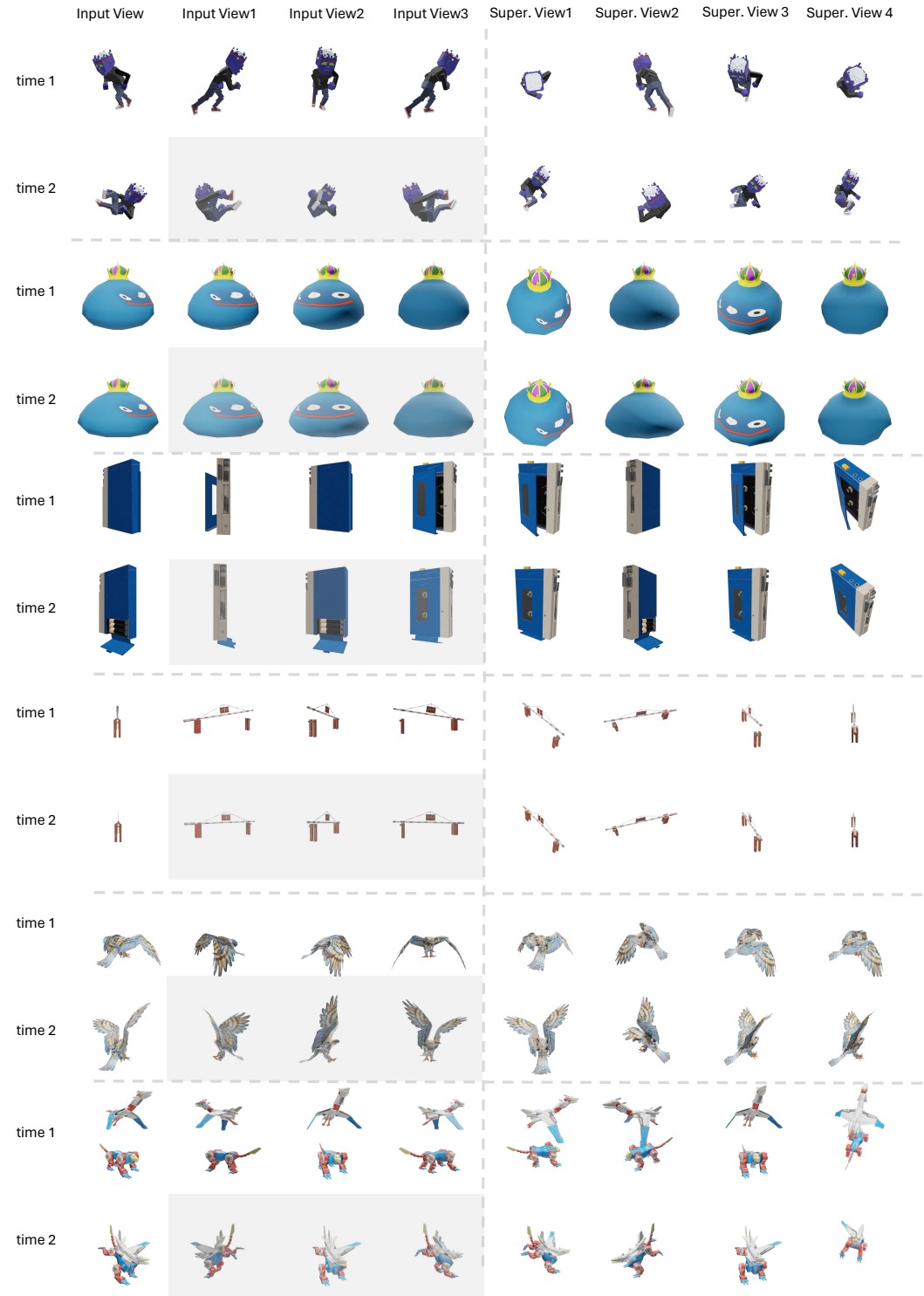

Figure 7: **Example training data.** Masked input views will not be visible to the model. They will be replaced by the copy of the multiview images at time $t = 1$.

Table 4: Comparison between L4GM and state-of-the-arts approaches on full metrics in the Consistent4D benchmark. Baseline results are from Gao et al. [15].

| Method | Pistol | | Guppie | | Crocodile | | Monster | | Skull | | Trump | | Aurorus | | Mean | |
|---|---|---|---|---|---|---|---|---|---|---|---|---|---|---|---|---|
| | LPIPS↓ | CLIP↑ | LPIPS↓ | CLIP↑ | LPIPS↓ | CLIP↑ | LPIPS↓ | CLIP↑ | LPIPS↓ | CLIP↑ | LPIPS↓ | CLIP↑ | LPIPS↓ | CLIP↑ | LPIPS↓ | CLIP↑ |
| D-NeRF [36] | 0.52 | 0.66 | 0.32 | 0.76 | 0.54 | 0.61 | 0.52 | 0.79 | 0.53 | 0.72 | 0.55 | 0.60 | 0.56 | 0.66 | 0.51 | 0.68 |
| K-planes [13] | 0.40 | 0.74 | 0.29 | 0.75 | 0.19 | 0.75 | 0.47 | 0.73 | 0.41 | 0.72 | 0.51 | 0.66 | 0.37 | 0.67 | 0.38 | 0.72 |
| C4D [21] | 0.10 | 0.90 | 0.12 | 0.90 | 0.12 | 0.82 | 0.18 | 0.90 | 0.17 | 0.88 | 0.23 | 0.85 | 0.17 | 0.85 | 0.16 | 0.87 |
| DG4D [39] | 0.12 | 0.92 | 0.12 | 0.91 | 0.12 | 0.88 | 0.19 | 0.90 | 0.18 | 0.90 | 0.22 | 0.83 | 0.17 | 0.86 | 0.16 | 0.87 |
| GFlow [15] | 0.10 | 0.94 | 0.10 | 0.93 | **0.10** | **0.90** | 0.17 | 0.92 | 0.17 | 0.92 | 0.20 | 0.85 | 0.15 | 0.89 | 0.14 | 0.91 |
| Ours | **0.08** | **0.98** | 0.12 | **0.94** | 0.09 | 0.89 | **0.15** | 0.92 | **0.12** | **0.95** | 0.15 | **0.96** | 0.13 | 0.92 | **0.12** | **0.94** |

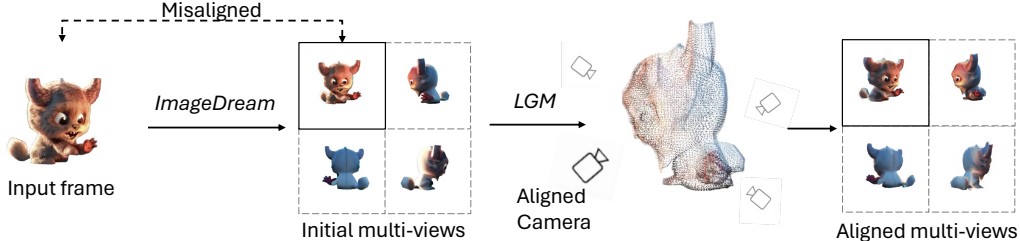

Figure 8: **Azimuth aligment.** ImageDream often generates multiviews that misalign with the input frame. We first use LGM to generate a 3D from the multiview images, then render it from different azimuths, and finally render it from the most-aligned azimuth and its other three orthogonal camera poses.

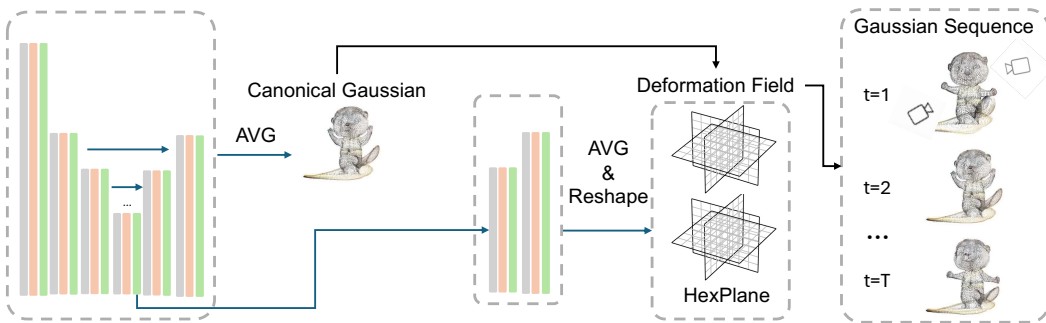

Figure 9: **HexPlane model.**

## E    Azimuth Alignment

After getting $P_{\text{init}}$, we render $P_{\text{init}}$ from a series of azimuth angles $\theta \in \{-180°, ..., 180°\}$ and select the azimuth that best aligns with the input frame. This is determined by $\theta_{\text{align}} = \text{argmin}_{\theta}||f(P_{\text{static}}, \theta) - I_1||_2^2$, where $f$ is a Gaussian volume rendering function. The final utilized sparse views, $\mathcal{J}_1$, are defined as $\{f(P_{\text{static}}, \theta_{\text{align}} + \Delta\theta)\}_{\Delta\theta \in \{0°, 90°, 180°, 270°\}}$. An illustration is shown in Figure 8.

## F    HexPlane Model Details

In this ablation study, we predict a canonical Gaussian and a deformation field represented by HexPlane. An additional decoder decodes the output from the U-Net middle block into 6 planes. The decoder is also equipped with cross-view and temporal attention. The canonical Gaussian then goes through the deformation field to produce $T$ sets of 3D Gaussians. The decoder has channels [1024, 128, 128]. We show an illustration in Figure 9.

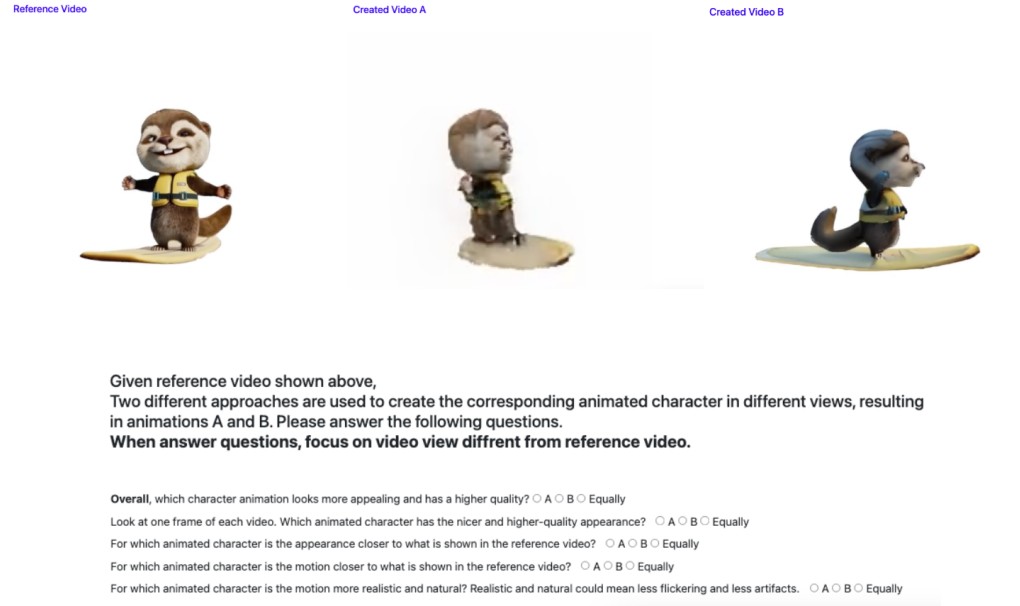

Figure 10: Screenshot of instructions provided to participants of the user studies for comparing L4GM and baselines.

# G More Qualitative Evaluation Details

## G.1 Evaluation Dataset

We collect 24 evaluation videos from multiple sources, including *1)* 10 videos from Emu video [16]; these are generated 4-second 16-FPS videos. *2)* 7 videos from Sora [34]; these are generated long 30-FPS videos. 3) 5 videos from Veo [10]; these are generated long 30-FPS videos. 3) 2 videos from ActivityNet [12]; these are real-world long videos of weight lifting.

We segment the foreground object with SAM-Track [8] and then crop and rescale the video to 512x512. Since the optimization-based baseline methods are very memory intensive, we trim all videos to 4 seconds and downsample all 30 FPS videos to 15 FPS, so that all videos have 64 frames.

## G.2 User Study

We conducted human evaluations (user studies) through Amazon Mechanical Turk to assess the quality of our generated 4D scenes, comparing them with DG4D [39], OpenLRM [19], STAG4D [67] and performing ablation studies.

We used the 24 examples evaluation dataset as described in Section 6.2. We rendered both baselines and our dynamic 4D scenes from similar camera perspectives and created similar videos. We first asked the participants to watch the reference input video and then asked them to compare the two videos with respect to 5 different evaluation axes and indicate a preference for one of the methods with an option to vote for 'equally good' in a non-forced-choice format. The 5 categories measure overall quality, 3D appearance quality, as well as motion alignment to reference video and motion realism.

For a visual reference, see 10 for a screenshot of the evaluation interface. In all user studies, the order of video pairs (A-B) was randomized for each question. In all user studies, each video pair was evaluated by five participants, totaling 120 responses for each of the baseline comparisons. We selected participants based on specific criteria: they had to be from English-speaking countries, have an acceptance rate above 95%, and have completed over 1000 approved tasks on the platform.

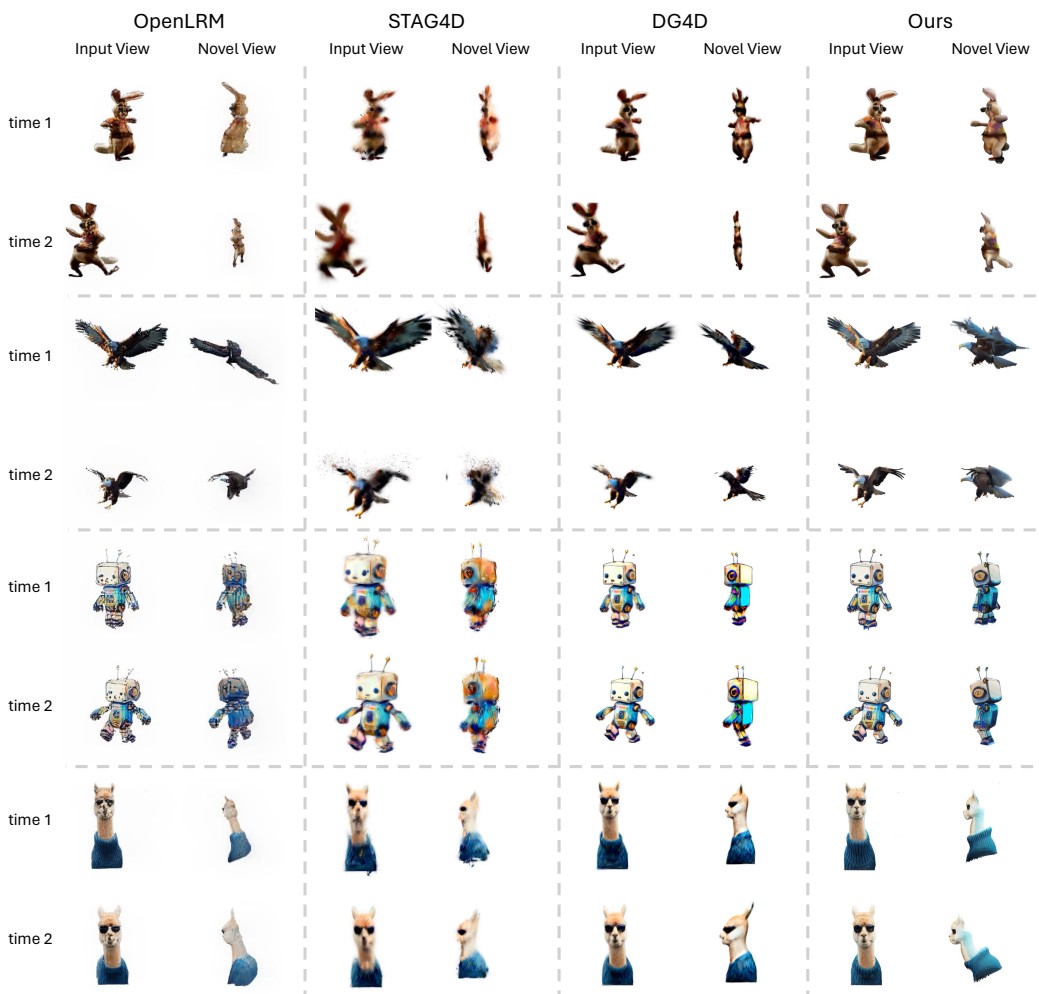

Figure 11: **More qualitative comparison** on generated videos.

## H More Qualitative Results

### H.1 More comparisons with state-of-the-art approaches

We show more qualitative comparisons with state-of-the-art approaches on both generated (Figure 11) and real-world (Figure 12) videos.

### H.2 Qualitative Results on Consistent4D Data

We show more qualitative results on the Consistent4D dataset in Figure 13.

## I Limitations

We show some limitations of L4GM in Figure 14. Our model can not handle motion ambiguity well. For example, in some walking motions, the model can successfully align with the reference view but the leg motion is not natural from other views. The model also cannot reconstruct multiple objects well, particularly when they occlude each other. Finally, the model fails to reconstruct objects from an ego-centric viewpoint, since the model assumes input views to be taken from $0°$ elevation.

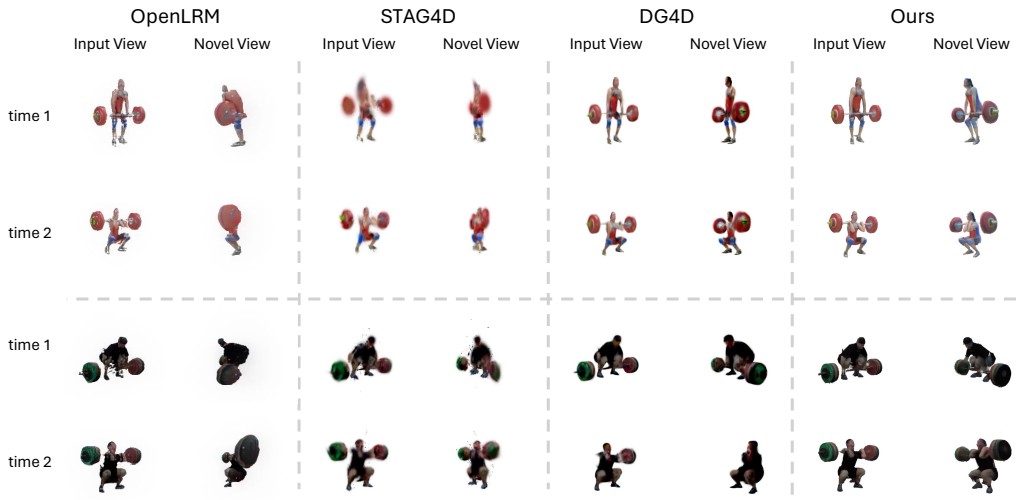

Figure 12: **More qualitative comparison** on real-world videos.

## J  More discussions

**Ablation results on the Consistent4D benchmark.**   In Table 5, we evaluate ablation models and interpolation models on the Consistent4D. Note that the quantitative results can be noisy since the benchmark only has 7 test samples. Comparing b) to a), we confirm that the pretrain plays a critical role. Comparing f) to a), we observe that the 4D representation is crucial. To evaluate the 4D interpolation model, we downsample the evaluation videos' framerate by 3, reconstruct downsampled videos with a), and then interpolate the 4D reconstruction with our interpolation model. Notably, g) achieves an on-par performance to a), which suggests the effectiveness of the interpolation model.

**Full attention.**   We explore replacing the temporal attention in L4GM with full attention. Full attention computes the self-attention on all images regardless of time and view, while temporal attention only computes the self-attention on images under the same viewpoint. As a result, full attention requires more computing. We show a memory-time analysis on a training iteration in Table 6, implemented with Efficient Attention. c) requires a larger memory usage and a longer processing time than b). In Figure 15, We further show a PSNR plot that compares temporal attention with full attention, trained on the GObjaverse subset. They achieve almost identical PSNR curves. Considering both the computation cost and empirical results, temporal attention would be a better design choice.

Table 5: **Ablation results on the Consistent4D benchmark.**

| Method | LPIPS↓ | CLIP↑ | FVD↓ |
|---|---|---|---|
| a) baseline | 0.11808 | 0.94037 | 652.81 |
| b) no pretrain | 0.13702 | 0.89055 | 851.62 |
| c) - temp attn. | 0.11792 | 0.93776 | 650.27 |
| d) + timeemb | 0.11888 | 0.93888 | 659.09 |
| e) + freezelgm | 0.11840 | 0.94006 | 729.24 |
| f) + Hexplane | 0.12602 | 0.92639 | 950.65 |
| g) + interpolation | 0.11896 | 0.94001 | 622.86 |
| h) GObjaverse | 0.11895 | 0.93960 | 630.55 |

**Multi-view multi-step videos input.**   In Figure 16, we show a multi-view multi-step video generated by ImageDream, after azimuth alignment. Concretely, we input each individual frame of the reference video to the ImageDream model and generate a multiview image for each time step. The obtained multiview video lacks a temporal consistency. Since ImageDream is a probabilistic generative model, even the same image input could produce very different multi-view generation. Therefore, we believe that such temporally inconsistent multi-view videos are not ideal input to the reconstruction model.

Table 6: **Attention.**

| Method | memory (Gb)↓ | time (s)↓ |
|---|---|---|
| a) cross view | 51.39 | 8.53 |
| b) cross-view + temp. | 53.83 | 9.64 |
| c) cross-view + full | 55.27 | 11.82 |
| d) full | 52.96 | 11.28 |

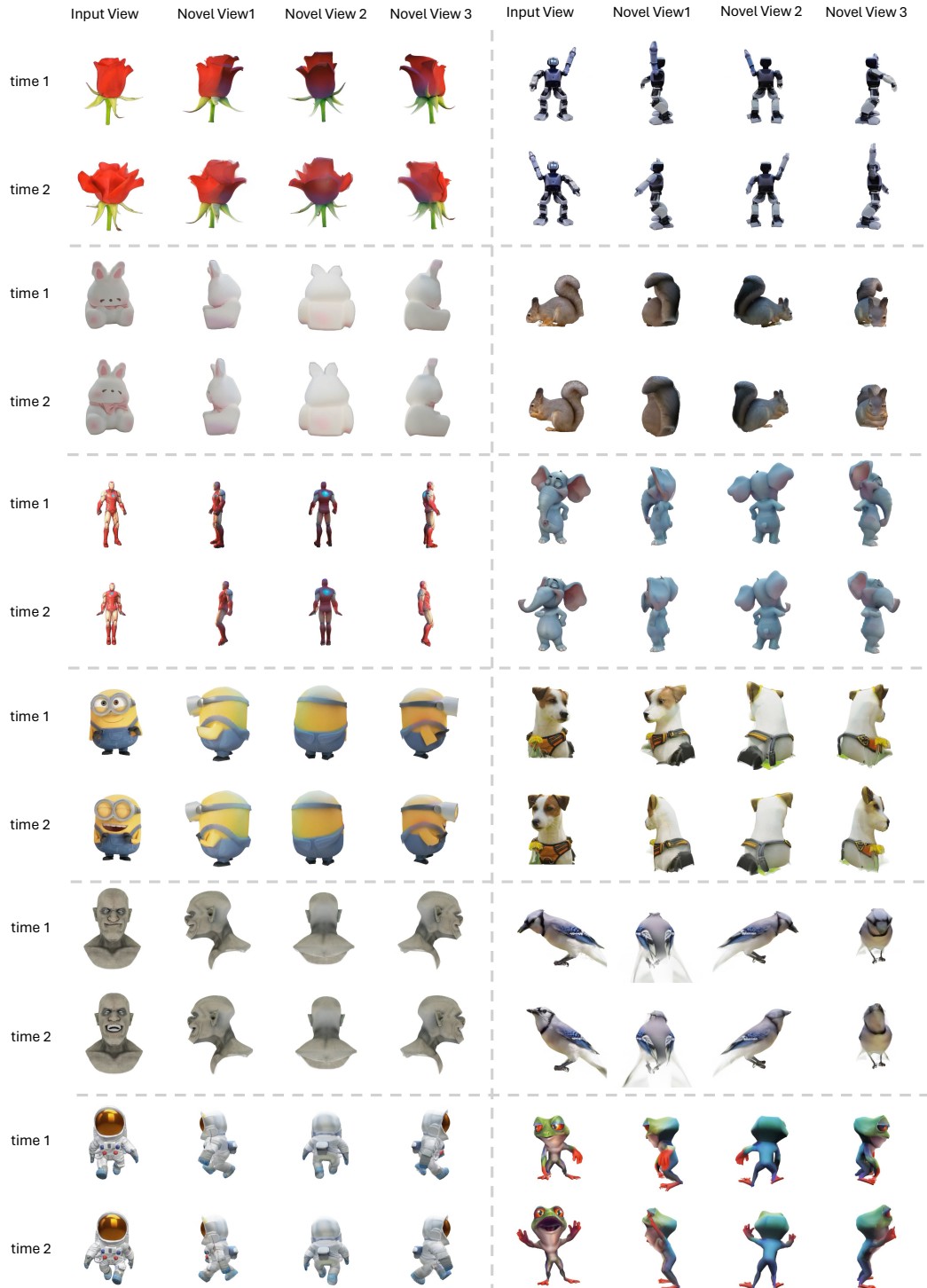

Figure 13: **Qualitative results on the Consistent4D dataset.**

**New evaluation on the GObjaverse subset** We provide quantitative results using the GObjaverse subset as training data in Table 5 line h. We manually verified that these Consistent4D testing samples are not part of the GObjaverse subset. The new results remain state-of-the-art and show no significant difference from the numbers reported in the main paper.

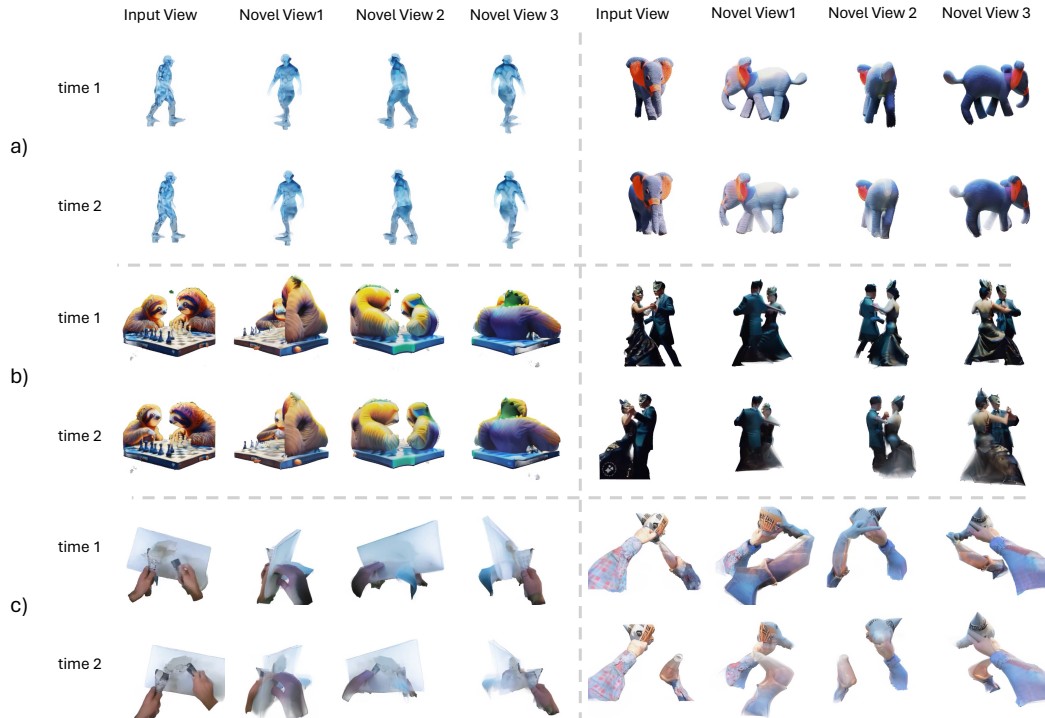

Figure 14: **Failure cases.** *a)* Motion ambiguity. *b)* Multiple objects with occlusions. *c)* Ego-centric videos taken from Ego4D [17].

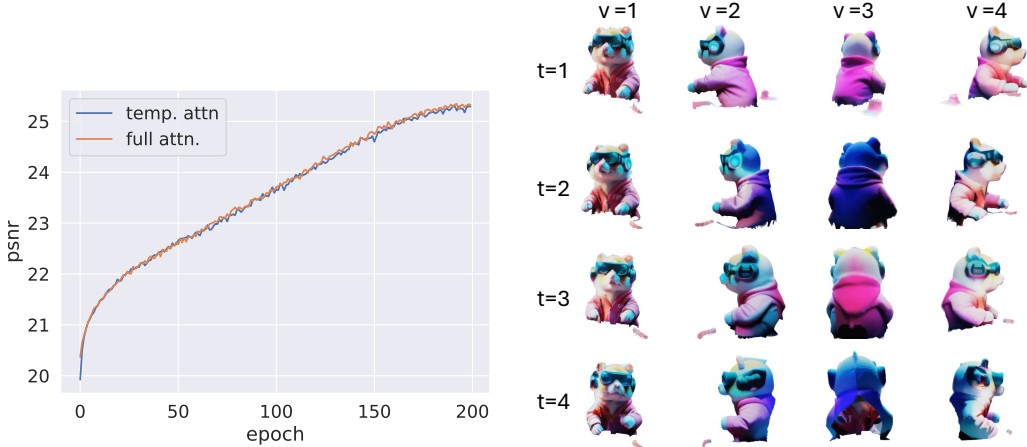

Figure 15: **Full-attention PSNR plot.**

Figure 16: **Multi-step multi-views.**

