# OpenReview forum: "L4GM: Large 4D Gaussian Reconstruction Model"
_NeurIPS.cc/2024/Conference — NeurIPS 2024 poster_

### Official Review · Reviewer_2k2z · 2024-07-08

**Soundness:** 3
**Presentation:** 4
**Contribution:** 2
**Rating:** 5
**Confidence:** 4

**Summary:**

This paper proposed L4GM, an efficient large 4D Gaussian reconstruction model to produce animated objects from videos by a single feed-forward. L4GM leverages ImageDream and LGM to achieve multiview images of the first frame as the input. The overall model is built upon the pre-trained LGM with cross-view and temporal attention blocks, which could be easily modified by rearranging the feature dimension. Besides, this paper proposed autoregressive reconstruction and 4D interpolation, largely improving the overall smoothness of the generated 4D. To train L4GM, the authors rendered animated objects from Objaverse, and conducted sufficient experiments to verify the effectiveness.

**Strengths:**

1. As a feed-forward technique, L4GM enjoys good efficiency and performance, especially for the high-resolution 4D generation.
2. The overall pipeline is convincing while the paper is also clearly written.
3. This paper contributes to large-scale 4D data with 12M videos rendered from Objaverse.

**Weaknesses:**

1. The main concern is the novelty. Although the overall pipeline is convincing, this paper is more like an extension of LGM for 4D generation. Most techniques are very straight-forward, such as temporal and cross-view attention, and multi-view synthesis by ImageDream+LGM.

2. Another concern is the setting of repeating the multiview images from the initial timestep as inputs for other frames. It works more like seeking temporary relief rather than a solid solution. For example, this repeating suffers from solving 4D generation with the motion of turning around, whose multiview images would contain conflict to the inputs. Why not use ImageDream+LGM to achieve more multiview inputs? Moreover, more quantitative and qualitative ablation studies should be considered for this setting, such as no-repeating views, repeating views, and multi-timestep-multi-view inputs respectively.

**Questions:**

1. Maybe there is a typo of temp. embed. in Figure 6(b), which should be time embed?
2. Why there are 12M videos at all? The authors said that 110k animations are captured with 48 views. So the overall videos should be about 5M.
3. Since L4GM is trained with animated objects from Objaverse,  the comparison of Consistent4D in Table 1 is not convincing enough. To the best of my knowledge, most objects in the Consistent4D test set are from Objaverse too. Please refer to https://github.com/yanqinJiang/Consistent4D/blob/main/test_dataset_uid.txt for more details.

**Limitations:**

The authors have discussed limitations in the supplementary.

---

> ### Author Rebuttal · Authors · 2024-08-07
>
> > Q1. The main concern is the novelty. Although the overall pipeline is convincing, this paper is more like an extension of LGM for 4D generation. Most techniques are very straight-forward, such as temporal and cross-view attention, and multi-view synthesis by ImageDream+LGM.
>
> A1. As pointed out by reviewers (4G9y, UkTx, hQSp), we would like to emphasize that L4GM is the first 4D reconstruction network. To achieve this, we propose several methods to successfully generate 4D assets in this setting. One of our core technical contributions is the novel framework itself. Regarding the reviewers’ concern about the techniques being "very straightforward," as mentioned in the main paper, we intentionally kept the model simple to ensure better scalability.
>
> We would also like to highlight that building a large 4D object dataset and training a reconstruction model on it have not been explored before. While the design choices may seem straightforward in hindsight, several uncertainties existed during the model's development. These included: 1) whether a model could learn to reconstruct dynamic 3D objects over time from a single multi-view, 2) identifying an appropriate input format for combining single-view video and multi-view images, and 3) determining a suitable 4D representation for the feed-forward model to predict.
>
> Thus, we believe L4GM is a major step towards generating high-quality 4D assets.
>
>
> > Q2. Another concern is the setting of repeating the multiview images from the initial timestep as inputs for other frames. It works more like seeking temporary relief rather than a solid solution.
> For example, this repeating suffers from solving 4D generation with the motion of turning around, whose multiview images would contain conflict to the inputs. Why not use ImageDream+LGM to achieve more multiview inputs? Moreover, more quantitative and qualitative ablation studies should be considered for this setting, such as no-repeating views, repeating views, and multi-timestep-multi-view inputs respectively.
>
> A2.  “Multi-timestep-multi-view" setting: Using ImageDream to generate more multiview inputs will generate very different 3D objects in different time steps. Please refer to an example in the rebuttal PDF. Generating temporally consistent multi-view videos is in fact a challenging task at the early stage of research [A].
>
> “No repeating views and repeating views” setting: We would like to reiterate that repeating views are only designed to let the model better handle the “T+V” input. Thus, no repeating views means that we need to change the model architecture to take a different input format, which we deem non-trivial and with the risk of hurting the pretrain weights.
>
> We agree that there are cases when the generated multiview images would conflict with the input video. We believe that the challenge can be tackled by improving the model's robustness to inaccurate multiviews, with strategies like augmentation and random dropout, which we leave for future works.
>
> [A] Liang et. al., Diffusion4D: Fast Spatial-temporal Consistent 4D Generation via Video Diffusion Models, 2024
>
> > Q3. Maybe there is a typo of temp. embed. in Figure 6(b), which should be time embed?
>
> A3. Yes, it is a typo. Thanks for pointing this out.
>
> > Q4. Why there are 12M videos at all? The authors said that 110k animations are captured with 48 views. So the overall videos should be about 5M.
>
> A4. One animation could be longer than one second. The 110k animations have 250k seconds after filtering, which sums up to 250k*48 = 12M 1-second videos.
>
> > Q5. Since L4GM is trained with animated objects from Objaverse, the comparison of Consistent4D in Table 1 is not convincing enough. To the best of my knowledge, most objects in the Consistent4D test set are from Objaverse too. Please refer to https://github.com/yanqinJiang/Consistent4D/blob/main/test_dataset_uid.txt for more details.
>
> A5. Thanks for pointing this out. We refer the reviewer to a new evaluation of the model only trained on the GObjaverse subset in the rebuttal PDF. We manually checked that these test samples are not part of the GObjaverse subset, so the issue can be avoided. The new results remain state-of-the-art and have no significant difference from the numbers we report in the main paper.

---

> > ### Comment · Reviewer_2k2z · 2024-08-10
> > **Thanks for the rebuttal**
> >
> > Thanks for the rebuttal. The rebuttal addressed most of my concerns. I raise my score to 5. Importantly, the limitation and related discussion about repeating views should be included in the revision.

---

> > > ### Author Response · Authors · 2024-08-12
> > > **Thanks to Reviewer 2k2z**
> > >
> > > Thank you for raising the score. We appreciate your suggestions and will include them in the revision.

---

### Official Review · Reviewer_hQSp · 2024-07-09

**Soundness:** 3
**Presentation:** 3
**Contribution:** 3
**Rating:** 6
**Confidence:** 4

**Summary:**

This paper utilizes rendering of animated objects from objverse(-xl) to extend lgm into 4D generation. Specifically, L4GM uses four orthogonal images of an object and the object's monocular dynamic video to obtain 3D Gaussians at each moment, enhancing the consistency between different moments through temporal self-attention layers. Subsequently, the smoothness of generated actions is further improved through a 4D Interpolation Model. L4GM leads in generation speed and evaluation metrics in the task of video-to-4D.

**Strengths:**

1. This paper significantly improves the generation speed of text-to-4D by utilizing a large-scale, object-centered dynamic dataset to extend lgm into 4D generation, and can model 4D objects in wild videos, achieving highly generalized results.
2. The paper proposes a 4D interpolation Model that can increase the frame rate of generated 4D objects, making the motion smoother. This will alleviate the problem of insufficient frame rates in video generation models.

**Weaknesses:**

The paper discusses extensively how to use dynamic datasets for pre-training, which is also a very important part of this work and will have a significant impact on the community. Whether this dataset is open source is also extremely important for evaluating this work, but the paper does not mention this point.

**Questions:**

1. In section 5, it is mentioned that some objects come with animations. Does this mean the objects already have predefined animations?
2. In the supplementary materials, I noticed that some generated 4D objects have limited motion range and motion rationality, which is a common issue in other 4D generation works. Since evaluating these is a resource-intensive task, are there any possible evaluation methods that could be discussed? This could be part of future work, and once an evaluation method is established, the motion quality of 4D generation could potentially be further improved.

**Limitations:**

The main issue lies in the amplitude and controllability of the animations. Is there a potential direction for this in the future?

---

> ### Author Rebuttal · Authors · 2024-08-07
>
> > Q1. The paper discusses extensively how to use dynamic datasets for pre-training, which is also a very important part of this work and will have a significant impact on the community. Whether this dataset is open source is also extremely important for evaluating this work, but the paper does not mention this point.
>
> A1. We will release the code for both the model and dataset building upon the acceptance of the paper and internal approval. Besides, we have tried our best to provide enough technical details in the paper to reproduce this work.
>
> > Q2. In section 5, it is mentioned that some objects come with animations. Does this mean the objects already have predefined animations?
>
> A2. Yes. The Objaverse dataset provides 3D objects with predefined 3D animations. Please kindly refer to this example in the Objaverse dataset https://skfb.ly/FWLt, where the dragon has been animated.
>
>
> > Q3. In the supplementary materials, I noticed that some generated 4D objects have limited motion range and motion rationality, which is a common issue in other 4D generation works. Since evaluating these is a resource-intensive task, are there any possible evaluation methods that could be discussed? This could be part of future work, and once an evaluation method is established, the motion quality of 4D generation could potentially be further improved.
>
> A3. In terms of "limited motion range and motion rationality", we highlight that we have performed a user study (in Table 2) on 24 videos from diverse sources (as detailed in Appendix G.1)  to evaluate the motion quality.  Our method achieves a high win rate over existing works on "Motion Realism" and "Motion Alignment w. Input Video". In addition, limited motions are not bound by our approach. As a reconstruction model, the reconstructed motion largely depends on the input video.
>
> We agree with the reviewer that a large-scale, comprehensive evaluation benchmark for 4D generation will be important to standardize and simplify evaluation. Although there exists a popular evaluation benchmark Consistent4D that automatically computes metrics, where we also evaluate our approach, the benchmark only provides 7 test samples and does not have a metric for motion quality. Improving the 4D evaluation benchmark is a crucial future work.
>
> > Q4. The main issue lies in the amplitude and controllability of the animations. Is there a potential direction for this in the future?
>
> A4. As mentioned above, the animation depends on the input video. Controlling the animations via video generation or editing is feasible but lies beyond the scope of this work. Our work only serves as a deterministic reconstruction model that reconstructs the animation from 2D to 3D.

---

> ### Comment · Reviewer_hQSp · 2024-08-08
>
> The author addressed my concerns. I agree that the animation is determined by the video. Thanks a lot. I will raise my score from borderline accept to weak accept.

---

> > ### Author Response · Authors · 2024-08-08
> > **Thanks to Reviewer hQSp**
> >
> > We are glad to hear that our response addressed your concerns. Thank you for raising the score.

---

### Official Review · Reviewer_UkTx · 2024-07-13

**Soundness:** 3
**Presentation:** 3
**Contribution:** 3
**Rating:** 6
**Confidence:** 3

**Summary:**

This work introduces a novel framework for generating animated 3D objects from single-view videos. The proposed framework employs a feed-forward approach, thus eliminating the need for computationally expensive optimization. The core idea is to create a large-scale synthetic multi-view video dataset and train a corresponding 4D LGM model, which is initialized from the 3D LGM [49] and enhanced with temporal self-attentions. Additionally, an interpolation network is introduced to improve the frame rate further. Overall, this method achieves superior performance in producing 4D assets, both in terms of efficiency and reconstruction quality.

**Strengths:**

- To the best of my knowledge, this is the first work to achieve feed-forward generation of 4D assets from a given video. The successful utilization of a large-scale dataset to tackle this challenging task may inspire further research in the community on related topics.
- The proposed framework demonstrates superior performance compared to existing methods across all evaluation metrics, as evidenced by the current evaluation (Figure 5, Tables 1). Additionally, the user study indicates a general preference for the results produced by this framework.
- Moreover, several significant ablation studies have been conducted on different components of the framework. Notably, the justification for using a pre-trained LGM (Figure 6(b)) is particularly convincing.
- Furthermore, the autoregressive generation method can produce animated objects with longer intervals, beyond the training time intervals.

**Weaknesses:**

Technical contributions

- Despite its impressive performance, this work's technical contributions are somewhat limited. The two main contributions are temporal self-attention and the 4D interpolation network.
- The temporal self-attention can be seen as a direct extension of the multi-view self-attention proposed in [49].
- While the 4D interpolation network can speed up inference, its overall performance does not seem to have a substantial impact, as demonstrated in the supplementary material video.

Having said that, I agree that a straightforward solution to a new problem should be recognized, and the above weakness is relatively minor.

**Questions:**

Some Questions:

- Regarding the temporal self-attention layer, why is only the time dimension considered in the self-attention mechanism rather than incorporating both multi-view and temporal information (full attention)? An analysis focusing on efficiency and performance would be valuable to understand the rationale behind this design choice.
- Why is the ablation study performed using PSNR (a reconstruction-based metric) instead of the metrics provided in Table 1?
- It appears that only qualitative comparisons are offered for the interpolation network. Could the performance be evaluated using the metrics shown in Table 1?
- Given that this work is an original contribution to the feed-forward generation of 4D assets, the community would greatly benefit from the release of the dataset and code to enable the reproduction of the work. Are there any plans to release these resources?

**Limitations:**

- This work also necessitates a "synthetic" multi-view video dataset for training. According to the implementation details, generating this dataset takes approximately 3 days using 200 GPUs. While this is easier to obtain than a real dataset, it still demands significant computational resources. Including a more comprehensive discussion on this in the limitations section would be beneficial.
- The proposed framework is claimed to generalize extremely well to in-the-wild videos (Abstract). However, there are only a few samples provided (4 samples in Figures 1 & 4), and many of these are in an animated style, with the exception of the left example in Figure 4. This left example displays various artifacts on the produced objects, such as blurred arms. Given the current evidence, this claim appears to be inadequately supported, and it is recommended to either include more samples or lower the claim accordingly.

---

> ### Author Rebuttal · Authors · 2024-08-07
>
> > Q1. Technical contributions:
>
> A1.  We appreciate that the reviewer agrees that “a straightforward solution to a new problem should be recognized”.We would also like to highlight that L4GM is the first feed-forward 4D reconstruction model, which could open up more possibilities for generating high-quality 4D assets.
>
> With regard to detailed technical contributions, as the reviewer pointed out, our core technical contribution is “a novel framework for generating animated 3D objects from single-view videos.” In terms of new techniques required to build such a model, besides the ones mentioned in the review, there are also 1)  autoregressive reconstruction and 2) multi-view copying for input batching.
>
> > Q2. The temporal self-attention can be seen as a direct extension of the multi-view self-attention proposed in [49].
>
> A2. We agree. We choose temporal self-attention to keep the 3D reconstruction pre-train as much as possible and also to keep the model simple for better scalability.
>
> > Q3. While the 4D interpolation network can speed up inference, its overall performance does not seem to have a substantial impact, as demonstrated in the supplementary material video.
>
> A3. Besides inference speed, the interpolation model also helps to alleviate the autoregressive reconstruction error so that we can reconstruct longer videos. Considering a reconstruction model that reconstructs a 1-second 30-FPS video at a time and can maximally self-reconstruct 10 times without quality drop, then the longest video to reconstruct will be 10 seconds. Our 3x-upsample interpolation model allows the reconstruction model to reconstruct a 3-second 10-FPS video at a time, thus improving the maximum video length from 10 seconds to 30 seconds.
>
> > Q4. Regarding the temporal self-attention layer, why is only the time dimension considered in the self-attention mechanism rather than incorporating both multi-view and temporal information (full attention)? An analysis focusing on efficiency and performance would be valuable to understand the rationale behind this design choice.
>
> A4. We use the temporal attention inspired by video diffusion models [2,3], which have been verified to work well when extending image generation to video generation at a large scale. Nonetheless, We agree that full attention is also a reasonable design choice, so we perform an additional experiment comparing full attention to temporal attention and show the results in the rebuttal PDF. Full attention requires more computation but brings no visible improvement to the results.
>
> > Q5.Why is the ablation study performed using PSNR (a reconstruction-based metric) instead of the metrics provided in Table 1?
>
> A5. The Consistent4D evaluation benchmark only has 7 test samples so the numbers can be noisy, thus we choose to show PSNR plots. Nonetheless, we carry out evaluations on the Consistent4D benchmark and present results in the rebuttal PDF.
>
> > Q6. It appears that only qualitative comparisons are offered for the interpolation network. Could the performance be evaluated using the metrics shown in Table 1?
>
> A6. Thanks for the suggestion. We have additionally evaluated the interpolation network on the Consistent4D benchmark by downsampling the test video framerates by 3 times. Results are presented in the rebuttal PDF. The numbers achieved by the interpolated 4D reconstruction and the baseline 4D reconstruction have no significant difference.
>
> > Q7.Given that this work is an original contribution to the feed-forward generation of 4D assets, the community would greatly benefit from the release of the dataset and code to enable the reproduction of the work. Are there any plans to release these resources?
>
> A7. We will release the code upon the acceptance of the paper and internal approval.
>
> > Q8. This work also necessitates a "synthetic" multi-view video dataset for training. According to the implementation details, generating this dataset takes approximately 3 days using 200 GPUs. While this is easier to obtain than a real dataset, it still demands significant computational resources. Including a more comprehensive discussion on this in the limitations section would be beneficial.
>
> A8. Thanks for the suggestion, we will discuss the approach’s requirement on computational resources and improve our limitation section.
>
> > Q9. The proposed framework is claimed to generalize extremely well to in-the-wild videos (Abstract). However, there are only a few samples provided (4 samples in Figures 1 & 4), and many of these are in an animated style, with the exception of the left example in Figure 4. This left example displays various artifacts on the produced objects, such as blurred arms. Given the current evidence, this claim appears to be inadequately supported, and it is recommended to either include more samples or lower the claim accordingly.
>
> A9. We will modify the tone of the claim and make it more accurate.  By “in-the-wild” we meant to emphasize that the model generalizes on unseen data domains that are different from our synthetic training data, like generated videos from Sora or real-world videos ActivityNet.

---

> > ### Comment · Reviewer_UkTx · 2024-08-12
> >
> > Thank you very much for the additional experiments and comments. These have addressed my concerns. I am maintaining my positive stance towards the current manuscript.

---

> > > ### Author Response · Authors · 2024-08-12
> > > **Thanks to Reviewer UkTx**
> > >
> > > Thank you for the response. We are glad to have your concerns clarified.

---

### Official Review · Reviewer_4G9y · 2024-07-13

**Soundness:** 3
**Presentation:** 3
**Contribution:** 3
**Rating:** 6
**Confidence:** 5

**Summary:**

This paper proposes a model for 4D reconstruction from a single video, building upon dynamic 3D gaussians and LGM architecture [49] previously applied to static 3D scenes. By processing generated multi-view images (derived from the first frame using prior method) alongside the video, the model outputs 4D Gaussians to reconstruct the video dynamics. The model was trained on a synthetic multi-view video dataset (Objaverse) and shows qualitative generalization to real-world images. Based on the good performance and extensive ablation studies, I recommend accepting this paper.

**Strengths:**

- The paper is clearly written, with good figures, structure, and well-explained motivation of design decisions
- The task of 4D reconstruction is both timely and of significant interest to the community.
- Though the paper contains strong-worded claims (see below), the demonstrated results show strong 4D reconstruction capabilities
- Ablation studies highlight the benefits of pretraining and the chosen representation.

**Weaknesses:**

- Claiming that model generalises “extremely well” to in-the-wild lacks empirical support (apart from cherry-picked qualitative results) and likely not true due to training assumptions (e.g. masks, static camera at 0 degree elevation). One possible evaluation to substantiate the claim would be a comparison with other works such as HyperNeRF.
- Since the code is not available, the details about the architecture and training are not sufficient for reproducing experiments and should be expanded. E.g. I could not find how exactly 32 heldout views are sampled neither in supplementary nor in the main paper.
- Minor: The architecture and training are not new and relies heavily on prior techniques, including a multi-view image generator. This perhaps aligns work more closely with the computer vision and engineering community (e.g., CVPR)

**Questions:**

- Could you provide more information about various axis of failure cases and the generalization limits to in-the-wild videos?
- Could you provide more details about "grid distortion" mentioned on line 594? The referenced paper also lacks clarity on this.
- Given the use of 128 80GB A100 GPUs, could you provide details on training the model with fewer resources? (particularly memory use)

**Limitations:**

Authors have acknowledged the limitations, however, the claim about generalizing "extremely well" on in-the-wild videos is unsubstantated and likely not true due to training assumptions (e.g. masks, static camera at 0 degree elevation).

---

> ### Author Rebuttal · Authors · 2024-08-07
>
> > Q1. Claiming that model generalises “extremely well” to in-the-wild lacks empirical support (apart from cherry-picked qualitative results) and likely not true due to training assumptions (e.g. masks, static camera at 0 degree elevation). One possible evaluation to substantiate the claim would be a comparison with other works such as HyperNeRF.
>
> A1. We will modify the tone of the sentence. By “in-the-wild” we emphasize on unseen data domains that are different from our synthetic training data, like generated videos from Sora or real-world videos ActivityNet. We agree that the method may not handle some real-world settings like non-zero elevation angles or unsegmented videos very well. Although we have discussed some failure cases in Appendix I, we thank the reviewer for bringing up HyperNeRF and will consider using that example to strengthen the limitation sections.
>
> > Q2. Since the code is not available, the details about the architecture and training are not sufficient for reproducing experiments and should be expanded. E.g. I could not find how exactly 32 heldout views are sampled neither in supplementary nor in the main paper.
>
> A2. We will release the code upon the acceptance of the paper and internal approval. The mentioned details can actually be found in Appendix B. L574, “For random cameras, we select a random elevation from [-5◦, 60◦]...We set the camera radius to 1.5“, where random cameras are the 32 heldout views.
>
> > Q3. Minor: The architecture and training are not new and relies heavily on prior techniques, including a multi-view image generator. This perhaps aligns work more closely with the computer vision and engineering community (e.g., CVPR)
>
> A3. We believe that this work is closely related to a wide range of artificial intelligence fields, like AI content generation, and 3D perception/simulation in robotics, so it should be well-fitted into the NeurIPS community.
>
> > Q4. Could you provide more information about various axis of failure cases and the generalization limits to in-the-wild videos?
>
> A4. As mentioned in A2, a brief analysis has been provided in Appendix I, including motion ambiguity, multi-object scenes, and ego-centric scenes. We will expand the discussion to include more failure case analysis.
>
> > Q5. Could you provide more details about "grid distortion" mentioned on line 594? The referenced paper also lacks clarity on this.
>
> A5. It is a data augmentation technique from the reference paper LGM[49]. Grid distortion simulates 3D inconsistency by grid sampling an image with a distorted grid. This makes the model more robust to inconsistent multiview input images. Please find the detailed implementation at https://github.com/3DTopia/LGM/blob/main/core/utils.py#L63.
>
> > Q6. Given the use of 128 80GB A100 GPUs, could you provide details on training the model with fewer resources? (particularly memory use)
>
> A6. Various practical techniques could be applied to trade of longer training time for less memory consumption, for example, gradient accumulation. Our training takes 1 day on 128 GPUs, so the same model training can be finished on 32 GPUs in 4 days with gradient accumulation, which is also a reasonable training time.
>
> > Q7. Authors have acknowledged the limitations, however, the claim about generalizing "extremely well" on in-the-wild videos is unsubstantated and likely not true due to training assumptions (e.g. masks, static camera at 0 degree elevation).
>
> A7. We agree that such limitations exist and will make the claims in the paper more accurate.

---

> > ### Comment · Reviewer_4G9y · 2024-08-12
> >
> > Thank you for your response.

---

### Author Rebuttal · Authors · 2024-08-07

We thank reviewers for the encouragement and insightful feedback. We are glad that the reviewers found:
- (4G9y, UkTx, hQSp) This is the first work to achieve feed-forward generation of 4D assets from a given video. The task of 4D reconstruction is timely and of significant interest, demonstrating strong results and superior performance compared to existing methods .
- (4G9y, UkTx) Ablation studies effectively highlight the benefits of pretraining and the chosen representation, with significant insights on different components .
- (hQSp, 2k2z) The proposed 4D interpolation model increases the frame rate and smoothness of motion, extending the capabilities of L4GM in high-resolution 4D generation.
- (2k2z, hQSp) The paper contributes significantly to large-scale 4D data with 12M videos rendered from Objaverse, showcasing highly generalized results.

We refer the reviewers to the uploaded PDF file for additional experiments, including:
- (UkTx) Evaluating ablation models and interpolation models on the Consistent4D benchmark.
- (UkTx) Explore replacing temporal attention with full attention.
- (2k2z) Generating multi-view multi-step videos using ImageDream.
- (2k2z) New evaluation on the GObjaverse subset, which has no overlap with the Consistent4D benchmark.

In addition, we would like to address some common concerns, including :
- (4G9y, hQSp, UkTx) Code and data release: We will release the code upon paper acceptance and internal approval.
- (4G9y, UkTx) Claim on in-the-wild:  We will modify the tone of the claim. By “in-the-wild” we emphasize on unseen data domains that are different from our synthetic training data,

Please kindly let us know if you have any further questions.

---

### Decision · Program_Chairs · 2024-09-25

**Decision:**

Accept (poster)

**Comment:**

The work received positive feedback from reviews, highlighting the impressive results, clear writing and acknowledging that this is the first work presenting a feed forward approach for 4D reconstruction from video. Slight concerns remained with respect to technical novelty. All in all though, the reviewers believe that this paper is of value to the community, due to the currently relevant topic and state of the art results.

I agree and follow with my accept recommendation.

Congratulations to the authors!